# Cross-cultural adaptation and measurement properties of the French version of the Trinity Amputation and Prosthesis Experience Scales —Revised (TAPES-R)

**François Luthi**[1,2,3]☯*, **Caroline Praz**[1,2]☯, **Bertrand Léger**[1,2], **Aurélie Vouilloz**[2], **Christine Favre**[2], **Isabelle Loiret**[4], **Jean Paysant**[4], **Noel Martinet**[4], **Alain Lacraz**[5], **Domizio Suva**[5], **Jean Lambert**[6], **Olivier Borens**[7], **Christos Karatzios**[3], **Philippe Vuistiner**[1,2]

1 Institute for Research in Rehabilitation, Clinique Romande de Réadaptation Suva, Sion, Switzerland, 2 Department of Medical Research, Clinique Romande de Réadaptation Suva, Sion, Switzerland, 3 Department of Physical Medicine and Rehabilitation, Orthopaedic Hospital, Sion, Switzerland, 4 Department of Physical Medicine and Rehabilitation, Institut Régional de Réadaptation, Nancy, France, 5 Division of Orthopaedics and Trauma Surgery, Geneva University Hospital (HUG), Genève, Switzerland, 6 Department of Musculoskeletal Medicine, Lausanne University Hospital (CHUV), University of Lausanne, Lausanne, Switzerland, 7 Service of Orthopaedics and Traumatology, Lausanne University Hospital (CHUV), University of Lausanne, Lausanne, Switzerland

☯ These authors contributed equally to this work.
* francois.luthi@crr-suva.ch

**Data Availability Statement:** All relevant data are within the manuscript and its Supporting Information files.

## Abstract

### Background

The Trinity Amputation and Prosthesis Experience Scales—Revised (TAPES-R) is a self-administered questionnaire to measure multidimensional adjustment to a prosthetic limb. Our aim was to assess the validity and reliability of the French version of the TAPES-R (TAPES-R-F).

### Materials and methods

The cross-cultural adaptation was performed according to the recommendations. Factor analysis and Rasch analysis were also performed to allow comparison with the original English version. Construct validity was assessed by measuring the correlations between TAPES-R-F subscores and quality of life, pain, body image satisfaction, anxiety and depression. Internal consistency was measured with Cronbach's α. The standard error of measurement, smallest detectable change, Bland and Altman limits of agreement, and intraclass correlation were the measures of agreement and reliability.

### Results

No major difficulties were encountered throughout the trans-cultural adaptation process. The final version of the TAPES-R-F was well accepted and understood by the patients. According to the factor analysis, the satisfaction scale should be treated as a one-

**Funding:** This study was supported by a donation of the Loterie Suisse Romande (LORO). The funders had no role in study design, data collection and analysis, decision to publish, or preparation of the manuscript.

**Competing interests:** The authors have declared that no competing interests exist.

dimensional construct when used by French-speaking people and should not be separated into two separate subscales, functional and aesthetic, as is the case in the original English version. Our study confirmed that there is a strong relationship between biopsychosocial factors and adjustment to amputation. Cronbach's α > 0.8 for all the subscales. Reliability was good to excellent for all the subscales (ICCs between 0.61 and 0.89). The smallest detectable changes were 0.7, 0.8, 1.3, 0.4, and 1.8 (general adjustment, social adjustment, adjustment to limitation, activity restriction, and global satisfaction with the prosthesis).

## Conclusions

The TAPES-R-F is a valid and reliable instrument to assess multidimensional adjustment of French-speaking lower limb amputees. This questionnaire can be used for both clinical assessment and research purposes.

## Introduction

The incidence of amputation varies between countries and ranges between 1.2 and 4.4 per 10,000 inhabitants per year [1, 2]. In addition, it is estimated that the demographic changes and the increasing rate of diabetes, which is the main cause of lower limb amputations, might double the number of amputations by 2050 compared to 2005 [3], although some databases may suggest a reduction in the number of amputations, possibly explained by better patient care [2, 4]. Whatever the future may hold, amputation will remain a major medical concern for the coming years. Being amputated of a limb is also an individual major challenge to face. It requires physical, psychological, functional, and social adaptation [5]. The adjustment to amputation and to artificial limb is a complex, multifaceted and a life-long process. Several factors may affect it, for example the cause and level of amputation, the length of time living with the prosthesis and the degree of prosthetic use, the psychosocial factors, the prosthetic design, the phantom limb sensation and pain and stump pain [5, 6]. An insufficient or inadequate adjustment may lead to different issues, like depression, anxiety, low self-esteem or social isolation [7–10]. Adjustment to amputation is therefore highly related to quality of life of the person with a lower limb amputation and is indeed considered as a good predictor and even as the key determinant of quality of life in this specific population [5, 11, 12].

To assess the psychosocial processes involved in adjusting to amputation and the specific demands of wearing a prosthesis, Gallagher et al. proposed the Trinity Amputation and Prosthesis Experience Scales with its nine subscales [13], which was then simplified to get the Trinity Amputation and Prosthesis Experience Scales—Revised (TAPES-R) with only six subscales [14]. It is a self-administered questionnaire with a 33-item psychosocial scale. It evaluates the multidimensional adjustment to a prosthetic limb, assessing psychosocial factors (general adjustment, social adjustment and adjustment to limitation), activity restriction and satisfaction with the prosthesis (functional satisfaction and aesthetic satisfaction). The TAPES-R contains a second section that looks at experience of phantom limb pain, residual limb pain, and other medical problems not related to the amputation, general health and physical capabilities. Like in the original study of Gallagher et al. about the TAPES-R [14], this part will not be further investigated here. The TAPES-R was elaborated and validated in English [14]. To date, the only validated version of the TAPES-R in another language is in Arabic [15], as all previous translations in German [16], Persian [17] and Turkish [18] have been made from the first version of the TAPES [13]. The aim of the present study was to conduct a cross-cultural

adaptation of the TAPES-R into French, the 5th most widely spoken language in the world with 300 million French speakers [19] and to evaluate its psychometric properties.

## Methods

### Setting

This study took place in 4 centres: two were orthopaedic tertiary centres in Switzerland (Geneva University Hospital and Lausanne University Hospital) and two were rehabilitation centres in Switzerland (Clinique Romande de Réadapation, Sion) and in France (Institut Régional de Réadaptation, Nancy). The inclusion criteria were: (1) to have had a lower limb amputation, (2) that the amputation occurred for at least one year, (3) to live at home and (4) to speak French. Conversely, the inability to read or understand French, being a resident in a health or retirement home were the only exclusion criteria. Eligible patients first received an information letter and were asked to give a written consent. Their sociodemographic and medical information were then found in the medical record of the patients. Permission to use and translate the TAPES-R was obtained from Pamela Gallagher who developed the original version. The study was approved by the ethical committee of the canton du Valais (CCVEM 043/11) and conducted in accordance with the Declaration of Helsinki.

### The questionnaire TAPES-R

The psychosocial scale of the TAPES-R is a 33-item self-administered multidimensional questionnaire to assess the psychosocial adjustment to lower limb amputation and to the wearing of prosthesis. It is subdivided into three sections: (1) the psychosocial section which comprises three subscales consisting of five items each (general adjustment, social adjustment and adjustment to limitation; each item rated on a 1 to 4 scale from "strongly agree" to "strongly disagree", with high scores indicative of a positive adjustment); (2) the activity restriction section which consists of ten items (rated on a scale from 0 to 2 from "No, not limited at all" to "Yes, limited a lot", with high scores indicative of an activity restriction); and (3) the satisfaction with the prosthesis which comprises eight items subdivided into two subscales (aesthetic (3 items) and functional (5 items) satisfaction, each item rated on a 1 to 3 scale from "Not satisfied" to "Very satisfied", with high scores indicative of a satisfaction with the prosthesis). It results in six subscores. The three adjustment and the activity restriction scores are obtained by averaging the scores of all the items of the subscale (thus resulting in a 1 to 4 scores). The two satisfaction scores by summing them (resulting in a 0 to 9, or 0 to 15 score). High scores are respectively indicative of great adjustment, activity restriction and satisfaction with the prosthesis.

### Cross-cultural adaptation

The cross-cultural adaptation of TAPES-R was performed according to the recommendations of the American Academy of Orthopaedic Surgeons outcomes committee [20] and other references in the literature [21, 22]. An expert committee was set up including methodologists, health professionals who were experts in the field, language professionals, and the translators. First of all, two non-physician translators whose mother tongue was French, and who were fluent in English, translated independently the questionnaire forward from English into French. One of them was informed about the aims of the study and about the concept of the questionnaire, while the second translator (so-called naive translator) received only limited information. The discrepancies between the two versions were solved by the two translators under supervision of a methodologist not involved in the translation process to form a

synthesis of the two forward-translations. Back-translation from French into English of this last version was performed by two native English speakers who were fluent in French and who were both totally blind to the original version. These two (naive) translators were not informed about the study and not specifically trained in the medical domain. The aim of this back-translation was to check that the translated version reflects the same items content as the original. The final discrepancies and inconsistences were examined and solved during a consensus meeting with the expert committee and a pre-final French version of the TAPES-R was drawn up. This pre-final version was administered to 10 native French speakers with lower limb amputation (five in Switzerland (Sion) and five in France (Nancy)). They were asked to point out the difficulties they encountered by answering the questionnaire. The definitive French version (TAPES-R-F) was then validated by taking into account their comments during a new consensus meeting.

## Measurement properties

The measurement properties were restricted to a subset of items relevant to cross-cultural adaptations [23]: content validity, internal consistency, construct validity, reproducibility (agreement and reliability), and floor or ceiling effects. The criterion validity has not been valued in the absence of a gold standard, as well as responsiveness due to the cross-sectional design of this study (no follow-up). As it is also important to consider possible differences that could limit direct comparisons between nations and cultures [24], factor analysis and Rasch analysis were performed. Factor analysis allows assessing whether a set of items is measuring a single dimension. A Rasch rating-scale model investigates whether the rating scales are used in the expected manner, according to suggested criteria [25]. The comparison of these results with those of the original English version of the TAPES-R allows the assessment of the cross-cultural validity of the questionnaire. To carry out these different measures, a sample of at least 50 patients is generally considered sufficient [23].

## Content validity

The content validity of the TAPES-R-F was assessed by the health professionals and methodologists who were members of the expert committee. They verified that the concept measured in the French version was similar to the original English version and that the target population was comparable. Interpretability of the questionnaire was verified with the 10 patients who participated in trans-cultural adaptation and measuring the percentage of missing values for each item or unreliable answers in the whole sample [23].

## Internal consistency

Internal consistency evaluates the homogeneity between the items of a subscale to verify that they all evaluate the same concept. It was analysed by calculating the Cronbach's α coefficient for each subscale. Cronbach's α values range from 0 (no internal consistency) to 1 (perfect internal consistency). A value above 0.8 is considered as acceptable [26].

## Construct validity

The construct validity measures the extent to which the scores of a questionnaire are associated with other measures in a way that is consistent with the assumptions derived from the concept studied. The French validated versions of the following questionnaires were used for this purpose:

**The 36-Item Short Form Health Survey (SF-36).** The SF-36 is a self-administered questionnaire to assess the health status [27, 28]. It consists of eight medical concepts (general

health, physical functioning, mental health, role limitations—physical, role limitations—emotional, vitality, bodily pain, and social functioning) assessed with 36 items. The questionnaire is subdivided into two parts: the physical component summary (PCS) and the mental component summary (MCS); each one evaluated on a 0 to 100 scale. A lower score indicates a greater disability.

**The Brief Pain Inventory (BPI).** The BPI is a self-administered questionnaire with 11 items used to assess the intensity of pain and the interference of pain on daily functions [29, 30]. Four items measure pain intensity (current pain, average pain, worst pain, and least pain) using a 0 to 10 scale from "no pain" to "pain as bad as you can imagine". BPI pain severity is scored as the mean of the four intensity items. Seven items measure the level of interference with function caused by pain (interference with general activity, mood, walking ability, normal work, relations with other persons, sleep, and enjoyment of life) using a 0 to 10 scale from "no interference" to "complete interference". BPI pain interference is scored as the mean of the seven interference items.

**The Hospital Anxiety and Depression scale (HADs).** The HADs is a self-administered questionnaire commonly used to detect states of anxiety and depression among patients [31, 32]. It is a 14-item scale, with seven items relating to anxiety and seven items focusing on depression. All items are rated from 0 to 3 with a maximal total score of 21 points for both anxiety and depression. Higher scores indicate greater level of anxious and depressive symptoms. The threshold at which clinical depression or anxiety should be suspected is ≥8.

**The Amputee Body Image Scale (ABIS).** The ABIS is a 20-item self-administered questionnaire designed to measure the body image perception of the people with an amputation [7, 33, 34]. The responses to each item range from 1 to 5 (from "none of the time" to "all the time") with a global score from 20 to 100, with high scores indicating great body image *concerns*. The shortened version of the ABIS (ABIS-R) has 14 items rated on a 0 to 2 scale ("Never", "Hardly ever or sometimes" or "Most of the time or All the time"), with a total score ranging between 0 and 28. A high score indicates that the patient perceives a high body disturbance.

**The Prosthetic Profile of the Amputee (PPA).** The PPA is a self-administered questionnaire, which collects information about the frequency of wear and use of lower limb prosthesis and identifies the factors potentially related to prosthetic use [35, 36]. We selected five items of this questionnaire (dichotomized as a yes or no answer): "trouble with putting the prosthesis", "wearing the prosthesis all the time", "needing technical help (crutches) to walk with the prosthesis", "Already fallen with the prosthesis" and "relatives' acceptation of the amputation and the prosthesis" to see if they were associated with the subscales of the TAPES-R-F.

To assess construct validity, correlations (Pearson) between these questionnaires and the TAPES-R-F subscales were calculated after a visual inspection of the distribution of the data to ensure that the normality assumption could be assumed. According to Evans, a correlation is considered as very strong if r > 0.80, strong if r between 0.60 and 0.79, moderate if r between 0.40 and 0.59, weak if r is between 0.20 and 0.39 and very week if r is lower than 0.20 [37]. It was hypothesised that the adjustment to limitation and functional satisfaction rather represent the physical side (stronger correlation (>0.40) with physical component of SF-36 and BPI interference), while social adjustment and aesthetic satisfaction would be more related to psychological aspects (stronger correlation (>0.40) with mental component of SF-36, ABIS, HADs). Eventually, general adjustment and activity restriction would be more global parameters linked to both physical and psychological aspects. We expected weaker correlation (≤0.40) between adjustment to limitation, activity restriction and functional satisfaction and the psychological aspects (mental component of SF-36, ABIS, HADs) and between social adjustment and aesthetic satisfaction and the physical part of the questionnaires (physical component of

SF-36, BPI interference). However, correlations are possible or even expected knowing that the biological and psychosocial parameters influence each other. Additionally, the associations between the items of the PPA and the subscales of the TAPES-R-F were evaluated. We expected a significant association between "trouble with putting the prosthesis", "wearing the prosthesis all the time" and "needing technical help (crutches) to walk with the prosthesis" and activity restriction and functional satisfaction, and between "relatives' acceptation of the amputation and the prosthesis" and social adjustment. All the five items should be associated with general adjustment. Construct validity is considered acceptable if at least 75% of the results are in correspondence with our hypotheses, which, in line with what is described above, are therefore 46 in number [23]. 75% of correspondence would here correspond to 35/46.

## Reproducibility

The reproducibility was measured by two administrations of the questionnaire at a one-week interval. In such a short time span, no significant difference should occur for the different scales of the TAPES-R. Agreement and reliability were distinguished [23]. Agreement concerns the absolute measurement error. A high agreement (and small measurement error) is necessary to distinguish clinically important changes from measurement error. To evaluate the agreement, the standard error of measurement (SEM) was calculated. It equals the square root of the error variance of an ANOVA analysis, including systematic differences [23]. The SEM can be converted into the smallest detectable change ($SDC_{ind}$) as $SDC_{ind} = 1.96 * \sqrt{2} * SEM$. $SDC_{ind}$ is the smallest change that can be interpreted as a "real" change, in one individual. The SDC measurable in a group of n people ($SDC_{group}$) is obtained as $SDC_{group} = SDC_{ind} / \sqrt{n}$. The limits of agreement (LoA) described by Bland and Altman were also reported. The LoA is the mean change in scores of repeated measurements $\pm$ 1.96 $\times$ standard deviation of these changes.

The reliability is important for discriminative purpose to distinguish among patients, despite measurement error [23]. It indicates to which degree the patients can be distinguish from each other. The intraclass correlation coefficient (ICC, range 0.00–1.00) was used as indicator of the reliability. It was computed with a two-way random-effects model (ICC(2,1) according to Shrout and Fleiss [38] or ICC(A,1) according to McGraw and Wong [39]) as systematic differences are considered to be part of the measurement error. This model investigates absolute agreement based on a single measurement. According to Cicchhetti et al., the reliability can be considered as good when the ICC is at least 0.6 and as excellent if at least 0.75 [40].

## Floor and ceiling effects

The proportion of patients achieving the lowest or highest possible score was computed in order to detect any floor or ceiling effects. Floor or ceiling effects were defined as present if at least 15% of results reached the minimum or the maximum score [41].

## Statistical analysis

Normal based confidence intervals (CI) were computed using 500 bootstrap replications; the confidence level was set at 95%. All statistical analyses were performed on complete-case data using Stata 15.0 (StataCorp, College Station, TX, USA). Factor analysis was used to evaluate the dimensional structure of each of the three TAPES-R-F scales (psychosocial adjustment, activity restriction, and satisfaction with the prosthesis). Rasch rating-scale models were then performed on the sets of items corresponding to each of the five subscales. Results were compared to those from the original English version of the TAPES-R [14] and to the Arabic translation [15] as a cross-cultural validation.

## Results

### Population

A convenient sample of 99 patients with lower limb amputation gave their informed consent and was included. Sixty-two patients answered the TAPES-R twice (test-retest) within an interval of one week. The characteristics of the participants and the mean scores in the different questionnaires are summarized in Table 1.

### Content validity

No important difficulty was encountered during the translation process. Only a few terms of the pre-final version were problematic (Activity restriction' subscale: "hobbies" was first

**Table 1. Characteristics of the subjects and scores for the different questionnaires.**

| | | subjects for validation (n = 99) |
|---|---|---|
| **Variable** | **Possible values** | **Mean (sd) or n (%)** |
| **Age at amputation [years]** | | 57 (17) |
| **Duration since amputation [years] (median–iqr)** | | 5 (3–8) |
| **Gender** | **Male** | 71 (72%) |
| | **Female** | 28 (28%) |
| **Level of amputation** | **Hip disarticulation** | 4 (4%) |
| | **Transfemoral** | 20 (20%) |
| | **Through the knee** | 18 (18%) |
| | **Transtibial** | 46 (46%) |
| | **Foot** | 11 (11%) |
| **Aetiology** | **Vascular** | 54 (55%) |
| | **Traumatic** | 34 (34%) |
| | **Other** | 11 (11%) |
| **Diabetes** | **Yes** | 32 (32%) (if we only consider the vascular cases: 54%) |
| | **No** | 67 (68%) (if we only consider the vascular cases: 46%) |
| **TAPES-R-F General adjustment** | **1–4** | 3.1 (0.7) |
| **TAPES-R-F Social adjustment** | **1–4** | 3.2 (0.7) |
| **TAPES-R-F Adjustment to limitation** | **1–4** | 1.9 (0.7) |
| **TAPES-R-F Activity restriction** | **0–2** | 1.2 (0.5) |
| **TAPES-R-F Aesthetic satisfaction** | **3–9** | 6.5 (1.8) |
| **TAPES-R-F Functional satisfaction** | **5–15** | 10.3 (3.3) |
| **TAPES-R-F Global satisfaction** | **8–24** | 16.7 (4.7) |
| **BPI severity** | **0–10** | 2.8 (2.2) |
| **BPI interference** | **0–10** | 2.9 (2.4) |
| **SF36-PCS** | **0–100** | 40.4 (9.0) |
| **SF36-MCS** | **0–100** | 46.9 (11.9) |
| **HADs-A** | **0–21** | 7.0 (3.9) |
| **HADs-D** | **0–21** | 6.3 (4.2) |
| **ABIS** | **20–100** | 57.1 (16.9) |
| **ABIS-R** | **0–28** | 14.6 (3.7) |

TAPES-R-F: Trinity Amputation and Prosthesis Experience Scales—Revised-French Version; BPI: Brief Pain Inventory; SF-36: 36-Item Short Form Health Survey; HADs: Hospital Anxiety and Depression scale, ABIS: Amputee Body Image Scale; IQR: interquartile range; SD: standard deviation

translated into "hobby". This term was clear for the Swiss patients but not for the French patients, so we changed it into *"passe-temps"*. "Flights of stairs" was first translated into *"rampes d'escaliers"*, but it was not clear for the patients due to confusion with the staircase support handrail and we changed it into *"étages d'escaliers"*). These terms have been modified so that the final version was well accepted and understood by all the patients. No more major semantic or language difficulties were noted. The percentage of missing values was low, indicating a good understanding of the participants. It was less than 5% for each of the items, with the exception of seven missing values (7.07%) for the item vi ("satisfaction with the reliability of the prosthesis").

## Factor analysis

We first computed the Kaiser-Meyer-Olkin (KMO) measure to determine whether the data were suited for factor analysis (FA). With KMO = 0.84, 0.88, and 0.87 for the psychosocial adjustment, activity restriction, and satisfaction with the prosthesis scales, respectively, our sample size of n = 99 seems adequate. Moreover, the Bartlett's tests of sphericity were significant, which support the presence of correlations between the items. The KMO values of each individual item were all larger than 0.77, which is "meritorious" according to Kaiser [42].

Exploratory FA confirmed the existence of three factors in the psychosocial adjustment scale (eigenvalues of 5.09, 3.17, and 1.36) that explained 99.1% of the total variance. The unidimensionality of the activity restriction scale was also confirmed (the first factor explained 89.2% of variance, eigenvalue = 4.97). Nevertheless, regarding the satisfaction with the prosthesis, the FA would rather suggest one single dimension in the French version (eigenvalue = 5.06, 86.5% of variance explained), not differentiating between functional and aesthetic satisfaction. This unidimensionality has also been observed in the Arabic translation of the TAPES-R [15]. A confirmatory FA was then performed on the items of the psychosocial adjustment scale. Table 2 shows the factor loadings after a direct oblimin oblique-rotation. An oblique rotation was chosen since the factors are not supposed to be independent. We can observe that each item is clearly related to one single factor.

**Table 2. Oblimin oblique-rotated factor loadings for all TAPES-R-F items of the psychosocial adjustment scale.**

| | Item | Factors | | |
|---|---|---|---|---|
| **Psychosocial Adjustment** | | **1** | **2** | **3** |
| General Adjustment | 1 | 0.014 | **0.787** | 0.119 |
| | 2 | -0.019 | **0.833** | -0.057 |
| | 3 | 0.226 | **0.616** | -0.111 |
| | 4 | 0.233 | **0.552** | -0.256 |
| | 5 | -0.047 | **0.876** | 0.011 |
| Social Adjustment | 6 | **0.745** | 0.102 | 0.164 |
| | 7 | **0.814** | 0.003 | -0.058 |
| | 8 | **0.883** | -0.032 | -0.033 |
| | 9 | **0.858** | 0.006 | -0.064 |
| | 10 | **0.562** | 0.207 | 0.309 |
| Adjustment to Limitation | 11 | 0.141 | -0.045 | **0.796** |
| | 12 | 0.026 | -0.120 | **0.655** |
| | 13 | 0.001 | 0.048 | **0.820** |
| | 14 | -0.228 | 0.150 | **0.708** |
| | 15 | 0.007 | -0.094 | **0.846** |

## Rasch analysis

The rating-scale model on the general psychosocial adjustment did not comply with the set criteria for category functioning. The probability of using rating category 2 was never higher than the three other ratings (Fig 1a). Category probability curves illustrate satisfactory category functioning in all other dimensions (Fig 1b, 1c, 1d and 1e).

The original TAPES used five categories for the adjustment subscales that were reduced to four after a Rasch analysis [14]. The present result would rather suggest to further reduce the rating to three categories for the general adjustment subscale, but since the three subscales on psychosocial adjustment share the same section of the questionnaire, it seems wiser to leave all items as is. Moreover, due to the relative small sample size, it is recommended not to make any definitive decision based on the Rasch analysis [43].

## Internal consistency

The Cronbach's coefficients were as follows: 0.87 (bootstrap 95% CI [0.81; 0.93]) for general adjustment, 0.90 (bootstrap 95% CI [0.86; 0.93]) for social adjustment, 0.87 (bootstrap 95% CI [0.81; 0.93]) for adjustment to limitation, 0.90 (bootstrap 95% CI [0.87; 0.93]) for activity restriction, and 0.92 (bootstrap 95% CI [0.89; 0.95]) for global satisfaction. A coefficient > 0.8 is expected for a satisfying consistency [26]. It is the case for all the subscales of the TAPES-R-F. The internal consistency was similar to what was observed in the Arabic translation [15], where Cronbach's α were 0.89, 0.89, and 0.87 for psychosocial adjustment, activity restriction, and global satisfaction, respectively.

## Construct validity

The correlations between the subscales of the TAPES-R-F and the different other questionnaires are given in Table 3. The near-normal distribution of the different scores was considered appropriate to conduct Spearman correlations. According to Evans' classification, a correlation coefficient < 0.20 is very weak, between 0.20 to 0.39 is weak, between 0.40 to 0.59 is moderate, between 0.60 to 0.79 is strong and ≥ 0.80 is a very strong correlation [37]. Only strong and moderate associations are presented here.

**General adjustment** showed a strong correlation with mental factors (mental component (MCS) of SF-36 and depression (HADs)), but also a moderate correlation with physical or pain-related factors (pain interference (BPI), physical component (PCS) of SF-36). These associations reflect the global dimension of the general adjustment.

**Social adjustment** is moderately associated with mental health status (MCS of SF-36), anxiety level (HADs), and body image disruption (ABIS).

**Adjustment to limitation** showed a moderate correlation with body image disruption (ABIS).

**Activity restriction** presented moderate to strong correlations with physical and pain—related factors (physical component (PCS) of SF-36 and pain interference (BPI)) and with mental factors (mental component (MCS) of SF-36, depression level (HADs) and body image disruption (ABIS)).

**Global satisfaction with the prosthesis** was moderately correlated both with physical and pain-related factors (pain severity, pain interference and physical component (PCS) of SF-36) and mental factors (depression level (HADs)).

Associations between TAPES-R-F and items of the PPA are given in Table 4.

**General adjustment** was significantly associated with three items of the PPA: "put the prosthesis easily", "always wear the prosthesis" and "have relatives who completely accept the amputation and the prosthesis".

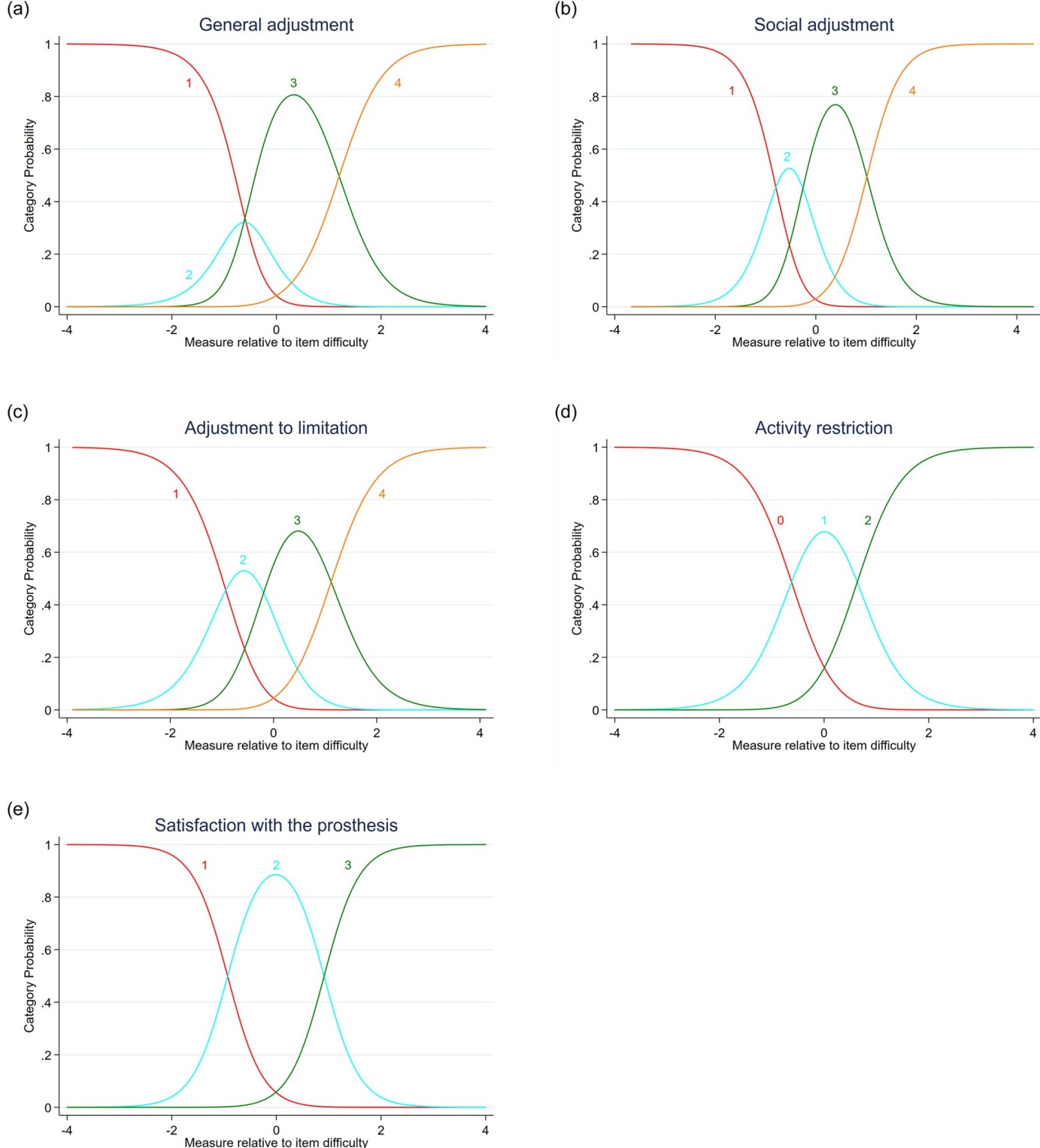

**Fig 1. a, b, c, d, e category probability curves of the different subscales of the TAPES-R-F.** The y-axis represents the probability of responding to one of the rating categories. The x-axis represents the different performance values.

**Table 3. Correlations and confidence intervals between the subscales of the TAPES-R-F and the subscales of BPI, SF-36 and HADs.**

| Subscales | BPI Severity | BPI Interference | SF-36_PCS | SF-36_MCS | HADs-A | HADs-D | ABIS |
|---|---|---|---|---|---|---|---|
| General adjustment | -0.38 [-0.54;-0.20] | -0.67 [-0.77;-0.55] | 0.46 [0.28;0.61] | 0.61 [0.46;0.73] | -0.46 [-0.60;-0.28] | -0.64 [-0.74;-0.50] | -0.55 [-0.68;-0.40] |
| Social adjustment | -0.25 [-0.43;-0.06] | -0.35 [-0.52;-0.16] | 0.23 [0.02;0.42] | 0.36 [0.16;0.53] | -0.40 [-0.56;-0.22] | -0.42 [-0.59;-0.23] | -0.50 [-0.64;-0.34] |
| Adjustment to limitation | -0.11 [-0.30;0.09] | -0.13 [-0.32;0.07] | 0.32 [0.12;0.50] | 0.04 [-0.17;0.25] | -0.02 [-0.22;0.18] | -0.27 [-0.44;-0.07] | -0.40 [-0.56;-0.22] |
| Activity restriction | 0.35 [0.17;0.51] | 0.46 [0.28;0.60] | -0.66 [-0.76;-0.52] | -0.42 [-0.58;-0.23] | 0.31 [0.11;0.48] | 0.56 [0.41;0.69] | 0.53 [0.37;0.66] |
| Aesthetic satisfaction | -0.24 [-0.42;-0.04] | -0.32 [-0.49;-0.12] | 0.36 [0.16;0.53] | 0.14 [-0.07;0.34] | -0.17 [0.36;0.03] | -0.24 [-0.42;-0.05] | -0.34 [-0.50;-0.15] |
| Functional satisfaction | -0.40 [-0.56;-0.22] | -0.52 [-0.65;-0.36] | 0.48 [0.30;0.62] | 0.34 [0.14;0.51] | -0.25 [-0.43;-0.05] | -0.41 [-0.56;-0.23] | -0.38 [-0.54;-0.20] |
| Global satisfaction | -0.38 [-0.54;-0.19] | -0.48 [-0.63;-0.31] | 0.44 [0.25;0.59] | 0.28 [0.08;0.47] | -0.29 [-0.46;-0.09] | -0.41 [-0.57;-0.23] | -0.47 [-0.61;-0.30] |

TAPES-R-F: Trinity Amputation and Prosthesis Experience Scales—Revised-French Version; BPI: Brief Pain Inventory; SF-36: 36-Item Short Form Health Survey; HADs: Hospital Anxiety and Depression scale, ABIS: Amputee Body Image Scale

**Social adjustment** was significantly associated only with the acceptance of the relatives and adjustment to limitation only with the item "put the prosthesis easily".

**Adjustment to limitation** was significantly associated with "put the prosthesis easily".

**Activity restriction** was significantly associated with three items of the PPA, namely "put the prosthesis easily", "need technical help to walk" and "have relatives who completely accept the amputation and the prosthesis".

**Global satisfaction** was significantly associated with "always wear the prosthesis" and "have relatives who completely accept the amputation and the prosthesis".

Thus, 34 out of 46 hypotheses (74%) were satisfied, which is very close to the recommendations with at least 75% of the assumptions confirmed [23].

### Reproducibility: Agreement and reliability

The SEM, $SDC_{ind}$ and the $SDC_{group}$ for the different subscales are presented in Table 5. The $SDC_{ind}$ corresponds to the smallest change in score that can be considered as above the measurement error, in an individual. The $SDC_{group}$ corresponds to the smallest change in score that can be considered as above the measurement error, in a group of 62 subjects.

Table 6 presents the mean differences in TAPES-R-F subscores between test and retest and the limits of agreement.

Reliability was good (between 0.6 and 0.74) to excellent ($\geq$ 0.75) for all the subscales, with ICCs between 0.61 and 0.89 (Table 6).

### Ceiling and floor effects

Considering that a ceiling or a floor effect is present if 15% of the respondents get the maximum or the minimum score, we observed a ceiling effect for two of the six subscales: general adjustment (17%), social adjustment (26%) and a floor effect for the adjustment to limitation (19%) (Table 7).

### Discussion

The purpose of this study was to make a cross-cultural adaptation of the TAPES-R into French and to measure its psychometric properties. No major difficulties were encountered throughout the process. The final version of the TAPES-R-F was well accepted and understood by the patients. The present results show that the TAPES-R-F is reliable and well suited to examine the psychosocial processes involved in adjusting to amputation and the specific demands of wearing prosthesis in French speaking people with a lower limb amputation. The factor

**Table 4. TAPES-R-F subscores in function of PPA answers and associations with p-values.**

| Variable | Possible values | n (%) | General adjustement | p-value | Social adjustement | p-value | Adjustement to limitation | p-value | Activity restriction | p-value | Global satisfaction | p-value | Aesthetic satisfaction | p-value | Functional satisfaction | p-value |
|---|---|---|---|---|---|---|---|---|---|---|---|---|---|---|---|---|
| Put the prosthesis easily | Yes | 83 (85.6%) | 3.2 (0.7) | <0.01* | 3.0 (0.7) | 0.21 | 2.0 (0.7) | <0.01* | 1.2 (0.5) | <0.01* | 17.6 (4.0) | 0.10 | 6.6 (1.8) | 0.88 | 10.7 (3.0) | 0.045* |
| | No | 14 (14.4%) | 2.7 (0.8) | | 3.2 (0.9) | | 1.4 (0.5) | | 1.7 (0.4) | | 15.7 (3.5) | | 6.5 (1.7) | | 8.9 (2.6) | |
| Always wear the prosthesis (at home and outside) | Yes | 35 (35.4%) | 3.4 (0.6) | <0.01* | 3.3 (0.7) | 0.58 | 1.8 (0.6) | 0.28 | 1.1 (0.4) | 0.13 | 18.8 (4.1) | <0.01* | 7.0 (1.6) | 0.06 | 11.7 (2.9) | <0.01* |
| | No | 64 (64.7%) | 2.9 (0.7) | | 3.3 (0.7) | | 2.0 (0.7) | | 1.3 (0.5) | | 16.4 (3.8) | | 6.2 (1.9) | | 9.4 (3.2) | |
| Already fallen with the prosthesis | Yes | 46 (52.3%) | 3.1 (0.7) | 0.13 | 3.1 (0.8) | 0.92 | 1.8 (0.5) | 0.14 | 1.2 (0.4) | 0.67 | 16.8 (4.2) | 0.21 | 6.4 (1.9) | 0.43 | 10.0 (3.3) | 0.08 |
| | No | 42 (47.7%) | 3.2 (0.6) | | 3.2 (0.6) | | 2.0 (0.8) | | 1.2 (0.5) | | 18.0 (3.9) | | 6.7 (1.6) | | 11.2 (2.7) | |
| Need technical help to walk | Yes | 67 (74.4%) | 3.1 (0.7) | 0.33 | 3.2 (0.8) | 0.74 | 1.9 (0.7) | 0.26 | 1.3 (0.5) | <0.01* | 17.2 (3.9) | 0.34 | 6.5 (1.8) | 0.40 | 11.1 (3.0) | 0.34 |
| | No | 23 (25.6%) | 3.3 (0.5) | | 3.2 (0.7) | | 2.0 (0.7) | | 1.0 (0.4) | | 18.1 (4.7) | | 6.8 (1.9) | | 10.4 (3.2) | |
| Complete acceptation (amputation and prosthesis) from the relatives | Yes | 43 (46.2%) | 3.3 (0.5) | 0.02* | 3.0 (0.6) | 0.03* | 2.0 (0.7) | 0.06 | 1.0 (0.5) | <0.01* | 18.5 (4.3) | <0.01* | 6.8 (1.8) | 0.30 | 11.7 (2.9) | <0.01* |
| | No | 50 (53.8%) | 3.0 (0.7) | | 3.4 (0.7) | | 1.8 (0.7) | | 1.4 (0.4) | | 16.1 (3.5) | | 6.4 (1.7) | | 9.2 (2.9) | |

TAPES-R-F: Trinity Amputation and Prosthesis Experience Scales—Revised -French Version; PPA: Prosthetic Profile of the Amputee

**Table 5. Standard error of measurement and smallest detectable change for individual and group for the different subscales of the TAPES-R-F.**

| Subscales | Standard error of measurement (SEM) | Smallest detectable change individual (SDC$_{ind}$) | Smallest detectable change group (SDC$_{group}$) |
|---|---|---|---|
| General adjustment | 0.27 | 0.73 | 0.09 |
| Social adjustment | 0.27 | 0.75 | 0.10 |
| Adjustment to limitation | 0.47 | 1.31 | 0.17 |
| Activity restriction | 0.16 | 0.44 | 0.06 |
| Aesthetic satisfaction | 0.98 | 2.73 | 0.35 |
| Functional satisfaction | 1.26 | 3.49 | 0.44 |
| Global satisfaction | 1.78 | 4.94 | 0.63 |

TAPES-R-F: Trinity Amputation and Prosthesis Experience Scales—Revised-French Version

analysis confirmed the existence of three psychosocial adjustment subscales, and one single subscale for activity restriction, as in the original English version, but unlike the Arabic version [15], where the psychosocial adjustment scale was considered as a single dimension. On the other hand, and similarly to what observed in the Arabic translation [15], our results show that only one global subscale for the satisfaction with the prosthesis in French-speaking people should be used, without treating aesthetic and functional satisfaction separately. This result could suggest that among people of French culture, at least in Switzerland and France, as well as among people of Arabic culture, aesthetic and functional satisfaction are not perceived as separate concepts, but rather as going with each other. In other words, it suggests that a prosthesis that is functionally satisfactory must also be aesthetically satisfactory and conversely. This result also underlines above all that the process of trans-cultural adaptation of a questionnaire does not consist only in a translation carried out according to the guidelines and in the measurement of usual psychometric qualities, but must also involve a more complete comparison with the concepts measured in the original version [24]. Researchers who would like to study satisfaction with the prosthesis in lower limb amputees using data from different cultures or countries should consider such possible cultural differences. If these differences are not taken into account, this can lead to interpretation bias [44]. The rating-scale diagnostics from the Rasch models also showed some disordered thresholds in the general adjustment subscale,

**Table 6. Mean difference in TAPES-R-F subscores between test and retest, the limits of agreement, ICC and 95% confidence interval.**

| Subscales | Mean difference | Limits of agreement | ICC | 95% CI |
|---|---|---|---|---|
| General adjustment | 0.12 | [-0.62; 0.85] | 0.79 | [0.67; 0.87] |
| Social adjustment | 0.07 | [-0.69; 0.82] | 0.82 | [0.72; 0.89] |
| Adjustment to limitation | 0.05 | [-1.26; 1.35] | 0.61 | [0.43; 0.75] |
| Activity restriction | 0.005 | [-0.44; 0.45] | 0.89 | [0.82; 0.93] |
| Aesthetic satisfaction | -0.11 | [-2.84; 2.62] | 0.64 | [0.47; 0.77] |
| Functional satisfaction | 0.34 | [-3.16; 3.83] | 0.81 | [0.70; 0.88] |
| Global satisfaction | 0.35 | [-4.59; 5.30] | 0.80 | [0.68; 0.87] |

TAPES-R-F: Trinity Amputation and Prosthesis Experience Scales—Revised-French Version; ICC: Intraclass Correlation Coefficient

**Table 7. Number of subjects who reached the minimum and the maximum values for the six subscales of the TAPES-R-F.**

| Subscale | Number of respondents with minimum value | Number of respondents with maximum value |
|---|---|---|
| General adjustment | 2 (2%) | 17 (17%) |
| Social adjustment | 0 (0%) | 26 (26%) |
| Adjustment to limitation | 19 (19%) | 1 (1%) |
| Activity restriction | 0 (0%) | 4 (4%) |
| Aesthetic satisfaction | 5 (5%) | 25 (25%) |
| Functional satisfaction | 4 (4%) | 14 (14%) |
| Global satisfaction | 2 (2%) | 12 (12%) |

TAPES-R-F: Trinity Amputation and Prosthesis Experience Scales—Revised-French Version

but confirmed the appropriateness of the rating categories for all other subscales. Since the three subscales on psychosocial adjustment share the same structure on the questionnaire, we suggest leaving it as is in the French version.

## Construct validity

Our study seems to confirm that there is a strong relationship between biopsychosocial factors and adjustment to amputation. Depressive symptoms, body image, and quality of life alterations have shown the strongest associations, what is consistent with the literature [45–47]. It also seems relevant to assess the psychological profile of the patient and his level of depressive symptoms. In a US national survey of people with lower limb amputations, Darnnal showed that 32.9% of people with significant depressive symptoms needed mental health services, but did not receive them [45]. The "complete acceptance from relatives" is another key parameter, since it is associated with four of the subscales, namely general adjustment, social adjustment, activity restriction, and satisfaction with the prosthesis. Relationships with relatives, which represents a significant part of social support, are indeed associated with the process of permanent adaptation that amputees experience on a daily basis [48, 49]. These findings should encourage health professionals to include early family members and close relatives into the rehabilitation process, explaining to them the situation and answering their questions in order to optimize acceptance by family members and close relatives and thus the patient's adaptation. Practical considerations must also be taken into account. The most important of them seems to be the ability to put the prosthesis easily. This should therefore be a major objective of the patient's therapeutic education [50], and it also underlines the role of the relationship between the patient and health professionals, in this case especially with the prosthetist [51]. Pain interference is moderately to strongly associated with three of the six subscales and also seems to play an important role in adjustment to amputation. Pain interference affects most people who have undergone amputation and has the potential to significantly affect participation in life activities [52].

## Internal consistency

Similar to the English version of the TAPES-R [14], the internal consistency is satisfying for all the subscales of the TAPES-R-F, which confirms that this questionnaire measures the concepts for which it was designed.

## Reproducibility: Agreement and reliability

The reliability was also good to excellent. Interestingly, no test-retest reliability was mentioned for the original TAPES or TAPES-R. In the validation of the Turkish and Persian versions of the TAPES [17, 18], the reliability was also acceptable for all the subscales. To our knowledge, it is the first measurement of the SEM and SDC for the TAPES-R. A change of 0.7, 0.8, 1.3, 0.4, and 4.9 for, respectively, general adjustment, social adjustment, adjustment to limitation, activity restriction, and global satisfaction can be considered as a "real" change in one individual (above the measurement error). Nevertheless, since no minimal clinically important difference (MCID) has been determined yet, it is difficult to interpret these values of SDC [53].

## Ceiling and floor effects

We detected ceiling or floor effect for 3 of the subscales. The ceiling effect was just above the 15% threshold for the general adjustment subscale (17%), but more pronounced for the social adjustment subscale (26%). It should therefore be kept in mind that participants with the maximum scores (i.e., with a good adaptation) cannot be distinguished from each other, with a reduced reliability as consequence, especially if we want to measure the social adjustment of people with lower limb amputations. Moreover, the responsiveness is limited because positive, respectively negative changes cannot be detected for these patients. For the original version, we do not have any analysis of ceiling or floor effect. Mazaheri et al. also showed ceiling effects for the Persian version [17], which would suggest that this observation depends on the questionnaire itself, but is not culturally specific. A floor effect was also measured for the adjustment of limitation (19%), which means that some participants with the lower limitations cannot be adequately detected with this subscale. Globally, these floor and ceiling effects indicate that the studied population has a very good psychosocial adaptation.

## Strengths and limitations

The TAPES-R has the great advantage of being a short questionnaire that requires little time either for the subject or for the investigator, while giving a global image of the fit to the amputation and to the prosthesis. The TAPES-R is also one of the recently recommended instruments for measuring the integration of amputees into the community [54]. The French version seems to be understandable and easily usable in the practice. However, the main limitation should be the sample of our patients that included people with a lower limb amputation living at home for several years, with a relatively low level of anxiety, depression, and pain. Due to the lack of data on people returning to live at home after amputation in Switzerland and France, we cannot know whether our sample is homogeneous enough to be truly representative of the whole population of amputees [55]. That can explain the scores showing a good adjustment that many of them reached for the different subscales. Additionally, in average, they underwent amputation for 5 (3–8) years (median—interquartile range (iqr)); they have had time to adjust well and to reach maximum scores. It would be worthwhile, in later studies, to use this questionnaire in different populations, e.g. with patients with a more recent amputation, ideally with a prospective design, or in another clinical setting like retirement home. Amputees living in such conditions are often alone with poor social support. Studies have explored a higher level of anxiety and depression in people with amputation who are alone [56, 57], which would likely be associated with poorer adaptation to amputation. Another possible limitation is that we have not been able to measure the influence of educational level. Only 48% of the participants answered to this question with compulsory schooling level in 66% of the respondents. Some comparisons with the original English version are also limited because the raw scores of the different scales have not been reported in the study [14].

## Conclusion

This study allowed the development of a French version of the TAPES-R with a cross-cultural adaptation and a validation (TAPES-R-F). The present study has demonstrated that the TAPES-R-F has satisfactory measurement properties comparable to those of the original English version. It is well understood and accepted by the patients and is a reliable instrument to assess the psychosocial adjustment, the activity restriction and the satisfaction with the prosthesis of French-speaking people with a lower limb amputation. However, our study suggests that satisfaction with the prosthesis measured by the TAPES-R-F should be treated as a single construct among French-native speakers, without separately assessing aesthetic and functional satisfaction. This questionnaire can be used both for clinical assessment and for research purposes even if it is necessary to remain attentive to a possible floor or ceiling effect in some people with lower limb amputation.

## Supporting information

**S1 Data.**
(PDF)

**S2 Data.**
(PDF)

**S3 Data.**
(XLSX)

## Acknowledgments

The authors thank Laure Huchon (Institut Régional de Réeadaptation, Nancy, France), and Khalid Segrouchni (Wallis Canton Hospital, Sion, Switzerland) for their active help in collecting the data, and all patients for their participation. The authors also thank Viviane Dufour (Institute for Research in Rehabilitation, Sion, Switzerland) for the preparation of questionnaires, mailings, and for help in formatting the database.

## Author Contributions

**Conceptualization:** François Luthi, Bertrand Léger, Aurélie Vouilloz, Christine Favre, Isabelle Loiret, Jean Paysant, Noel Martinet, Alain Lacraz, Jean Lambert, Philippe Vuistiner.

**Data curation:** Caroline Praz.

**Formal analysis:** François Luthi, Caroline Praz, Philippe Vuistiner.

**Funding acquisition:** François Luthi, Bertrand Léger.

**Investigation:** François Luthi, Caroline Praz, Aurélie Vouilloz, Christine Favre, Isabelle Loiret, Jean Paysant, Noel Martinet, Alain Lacraz, Domizio Suva, Jean Lambert, Olivier Borens, Christos Karatzios.

**Methodology:** François Luthi, Caroline Praz, Bertrand Léger, Philippe Vuistiner.

**Project administration:** François Luthi, Bertrand Léger.

**Resources:** Bertrand Léger.

**Supervision:** François Luthi, Caroline Praz, Bertrand Léger, Philippe Vuistiner.

**Validation:** François Luthi, Caroline Praz, Aurélie Vouilloz, Christine Favre, Isabelle Loiret, Jean Paysant, Noel Martinet, Alain Lacraz, Domizio Suva, Jean Lambert, Olivier Borens, Christos Karatzios, Philippe Vuistiner.

**Visualization:** François Luthi, Caroline Praz, Philippe Vuistiner.

**Writing – original draft:** François Luthi, Caroline Praz.

**Writing – review & editing:** François Luthi, Caroline Praz, Bertrand Léger, Aurélie Vouilloz, Christine Favre, Isabelle Loiret, Jean Paysant, Noel Martinet, Alain Lacraz, Domizio Suva, Jean Lambert, Olivier Borens, Christos Karatzios, Philippe Vuistiner.

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
