## [Decision Letter · Decision Letter 0]

4 Dec 2019

PONE-D-19-21400

Cross-cultural adaptation and measurement properties of the French version of the Trinity Amputation and Prosthesis Experience Scales - Revised (TAPES-R)

PLOS ONE

Dear Dr. Luthi,

Thank you for submitting your manuscript to PLOS ONE. After careful consideration, we feel that it has merit but does not fully meet PLOS ONE’s publication criteria as it currently stands. Therefore, we invite you to submit a revised version of the manuscript that addresses the points raised during the review process.

First of all, let me apologize for the delay in the review of your manuscript. Finding reviewers willing to evaluate this manuscript has been very hard. In any case, I was able to collect a review from just one reviewer, who recommends a minor revision of your study. Because the manuscript has been under review for too long, I think that a review could be undertaken at this point with the current study. However, and after my own reading of the manuscript, I would consider this as a major revision. 

Therefore, please notice that a resubmission will require an additional round of reviews, and that the final outcome of the process cannot be predicted at this point. If you decide to resubmit a revised version of your manuscript, please provide either a proper answer or rebuttal to each of the suggestions that were raised by the Reviewers.

We would appreciate receiving your revised manuscript by Jan 18 2020 11:59PM. To enhance the reproducibility of your results, we recommend that if applicable you deposit your laboratory protocols in protocols.io, where a protocol can be assigned its own identifier (DOI) such that it can be cited independently in the future. For instructions see: http://journals.plos.org/plosone/s/submission-guidelines#loc-laboratory-protocols

We look forward to receiving your revised manuscript.

Kind regards,

Angel Blanch, Ph.D.

Academic Editor

PLOS ONE

Journal Requirements:

2. Please include a separate caption for each figure in your manuscript.

Additional Editor Comments (if provided):

Reviewers' comments:

Reviewer's Responses to Questions

**Comments to the Author**

1. Is the manuscript technically sound, and do the data support the conclusions?

Reviewer #1: Yes

2. Has the statistical analysis been performed appropriately and rigorously? 

Reviewer #1: Yes

3. Have the authors made all data underlying the findings in their manuscript fully available?

Reviewer #1: Yes

4. Is the manuscript presented in an intelligible fashion and written in standard English?

Reviewer #1: Yes

5. Review Comments to the Author

Reviewer #1: Dear Authors,

I am grateful for the efforts to validate TAPES-R in the French language. The design of the research, collection and analysis of data, and presentation of the findings are straightforward and scientifically rigor. The steps of cultural adaptation and validation process are thoroughly followed by the authors. The reviewer acknowledges the practical relevance and research advancements of this research, which will allow many practitioners to measure the outcomes of their support and understand the overall wellbeing of the people affected by amputation.

The reviewer recommends publishing this research, after reviewing and adjusting a few methodological and practical comments outlined below.

1. To date, there is no validated version of the TAPES-R in another language, as all previous translations in German, Persian and Turkish have been made from the first version of the TAPES. (Page 11)

- Kind request to review a recent study of Massarweh and Sobuh (2019) for TAPES-R Arabic version and adjust accordingly.

2. Conversely, the inability to read or understand French, being a resident in a health or retirement home was the only exclusion criteria. (page 11)

- The reasons for excluding the participants from retirement homes should have been mentioned. Mostly the people living in retirement homes are single and without family/social support. The studies have explored a higher level of anxiety and depression in people with amputation who are alone (Parkes, 1976; Hawamdeh, Othman, & Ibrahim, 2008).

3. In any validation studies of a psychological tool, the sample size remains crucial. The authors could not explain the nature of the sample size, it seems, a convenience sample was used. The sample size determination techniques would have been better explained either in the main section or in limitations (Etikan, Musa, & Alkassim, 2016). For the factor analysis, a minimum of 6-10 samples per item of measure is required to draw complete factor loads and measurement fits of the data. As confirmatory factor analysis was not carried out, the EFA and Rasch analysis were presented. The results received from Rasch analysis, for less than 100 samples, have inadequate and sometimes opposite conclusions (Chen et al., 2014). Therefore, the reviewer recommends the authors to address the sample methods, sample size, and interrelated issues. Kaiser-Meyer-Olkin (KMO) sampling adequacy test results would have been best to present if possible.

4. The researchers are also requested to state/describe the normality test outcomes of the data, because the result section does not include it. However, the data analysis section of methodology outlines it.

5. All the five items should be associated with general adjustment. Construct validity is considered acceptable if at least 75% of the results are in correspondence with our hypotheses, which would here correspond to 34/46 (Page 17)

- The presentation of construct validity outcomes seems very vague, cannot be well understandable. The sentence, ‘Construct validity is considered acceptable if at least 75% of the results are in correspondence with our hypotheses, which would here correspond to 34/46’, needs further elaboration.

6. It would be better to provide factor loadings of all items of each measure, so that the readers will understand better (Page 21 under the factor analysis).

7. The present result would rather suggest to further reduce the rating to three categories for the general adjustment subscale, but leave it as is all other subscales. Nevertheless, since the three subscales on psychosocial adjustment share the same section of the questionnaire, it seems wiser to leave all items as is.

- I would have deleted the highlighted text in Yellow.

References:

Massarweh, R., & Sobuh, M. M. (2019). The Arabic Version of Trinity Amputation and Prosthetic Experience Scale-Revised (TAPES-R) for Lower Limb Amputees: Reliability and Validity. Disability, CBR & Inclusive Development, 30(1), 44-56. doi: 10.5463/dcid.v30i1.718

Gallagher, P., & MacLachlan, M. (2004). The Trinity Amputation and Prosthesis Experience Scales and quality of life in people with lower-limb amputation. Archives of Physical Medicine and Rehabilitation, 85(5), 730-736. doi: 10.1016/j.apmr.2003.07.009

Arafat, S. (2016). Cross cultural adaptation & psychometric validation of instruments: Step-wise description. International Journal of Psychiatry, 1(1), 1-4. doi: 10.33140/IJP/01/01/00001

Parkes, C. M. (1976). The psychological reaction to loss of a limb: the first year after amputation. In Modern perspectives in the psychiatric aspects of surgery (pp. 515-532). Palgrave Macmillan UK.

Hawamdeh, Z. M., Othman, Y. S., & Ibrahim, A. I. (2008). Assessment of anxiety and depression after lower limb amputation in Jordanian patients. Neuropsychiatric disease and treatment, 4(3), 627-633. doi: 10.2147/NDT.S2541

Etikan, I., Musa, S. A., & Alkassim, R. S. (2016). Comparison of convenience sampling and purposive sampling. American journal of theoretical and applied statistics, 5(1), 1-4. doi: 10.11648/j.ajtas.20160501.11

Chen, W. H., Lenderking, W., Jin, Y., Wyrwich, K. W., Gelhorn, H., & Revicki, D. A. (2014). Is Rasch model analysis applicable in small sample size pilot studies for assessing item characteristics? An example using PROMIS pain behavior item bank data. Quality of life research, 23(2), 485-493. doi: 10.1007/s11136-013-0487-5

6. PLOS authors have the option to publish the peer review history of their article (what does this mean?). If published, this will include your full peer review and any attached files.

Reviewer #1: Yes: Yubaraj Adhikari

---

## [Author Response · Author response to Decision Letter 0]

17 Jan 2020

Dear Professor Blanch,

We are pleased to send you a revised version of the manuscript entitled : “Cross-cultural adaptation and measurement properties of the French version of the Trinity Amputation and Prosthesis Experience Scales - Revised (TAPES-R)”, by François Luthi, Caroline Praz, Bertrand Léger, Aurélie Vouilloz, Christine Favre, Isabelle Loiret, Jean Paysant, Noel Martinet, Alain Lacraz, Domizio Suva, Jean Lambert, Olivier Borens, Christos Karatzios and Philippe Vuistiner, submitted for publication in PLOS ONE (PONE-D-19-21400).

We hope to have adequately addressed all the issues for this revision process and we think that the manuscript has been significantly improved. We are also grateful to the reviewer for their contribution. As requested, we have included a rebuttal letter that responds to each point raised by the academic editor and reviewers, as well as the manuscript with and without tracked changes (separate files).

Thank you for considering the submission of our revised manuscript for publication, and we look forward to hearing from you at your earliest convenience.

Sincerely,

François Luthi

---

## [Decision Letter · Decision Letter 1]

30 Jan 2020

Cross-cultural adaptation and measurement properties of the French version of the Trinity Amputation and Prosthesis Experience Scales - Revised (TAPES-R)

PONE-D-19-21400R1

Dear Dr. Luthi,

We are pleased to inform you that your manuscript has been judged scientifically suitable for publication and will be formally accepted for publication once it complies with all outstanding technical requirements.

With kind regards,

Angel Blanch, Ph.D.

Academic Editor

PLOS ONE

Additional Editor Comments (optional):

Reviewers' comments:

Reviewer's Responses to Questions

**Comments to the Author**

1. If the authors have adequately addressed your comments raised in a previous round of review and you feel that this manuscript is now acceptable for publication, you may indicate that here to bypass the “Comments to the Author” section, enter your conflict of interest statement in the “Confidential to Editor” section, and submit your "Accept" recommendation.

Reviewer #1: All comments have been addressed

2. Is the manuscript technically sound, and do the data support the conclusions?

Reviewer #1: Yes

3. Has the statistical analysis been performed appropriately and rigorously? 

Reviewer #1: Yes

4. Have the authors made all data underlying the findings in their manuscript fully available?

Reviewer #1: Yes

5. Is the manuscript presented in an intelligible fashion and written in standard English?

Reviewer #1: Yes

6. Review Comments to the Author

Reviewer #1: I am grateful for the authors who went through the comments and recommendations of the reviewer. All relevant recommendations were addressed in a scientific and statistically sound manner. I agree with the authors that the quality of the paper is improved. This article and tool deserve to be published so that the end-users can benefit from it.

During my over a decade long career in International Red Cross, I can assure, our colleagues from the physical rehabilitation programme unit can also benefit from this tool in measuring the psychosocial wellbeing of the amputated beneficiaries who receive technical assistance. After reviewing the revised text, I have no further comments to the authors. I would like to provide my gratitude to the authors and the board of editors for providing this opportunity.

7. PLOS authors have the option to publish the peer review history of their article (what does this mean?). If published, this will include your full peer review and any attached files.

Reviewer #1: Yes: Yubaraj Adhikari (Ph.D. Scholar at University of Nicosia, MSc in Mental Health Psychology from University of Liverpool)

---

## [Editor Report · Acceptance letter]

7 Feb 2020

PONE-D-19-21400R1 

Cross-cultural adaptation and measurement properties of the French version of the Trinity Amputation and Prosthesis Experience Scales - Revised (TAPES-R) 

Dear Dr. Luthi:

I am pleased to inform you that your manuscript has been deemed suitable for publication in PLOS ONE. Congratulations! Your manuscript is now with our production department. 

With kind regards,

on behalf of

Dr. Angel Blanch 

Academic Editor

PLOS ONE